# Proportions of Early Specializers Varies According to Methods and Skill Level

**DOI:** 10.3390/sports10030034

**Published:** 2022-02-28

**Authors:** Alexandra Mosher, Jessica Fraser-Thomas, Melissa J. Wilson, Joseph Baker

**Affiliations:** 1School of Kinesiology and Health Science, Faculty of Health, York University, Toronto, ON M3J 1P3, Canada; jft@yorku.ca (J.F.-T.); bakerj@yorku.ca (J.B.); 2Swimming Australia, Brisbane 4109, Australia; melissa.wilson@swimming.org.au

**Keywords:** early specialization, development, milestones, classification

## Abstract

Sport researchers have warned about the lack of a clear and consistent definition of early specialization, while others have raised concerns around the validity of methods used to classify athletes as ‘specializers’. The current investigation includes two studies examining the implications of varying classification methods for exploring both specialization and early specialization in sport. Study 1 examined whether different approaches to defining and measuring specialization affected the classification of athletes throughout development and provided a ‘profile’ of the sample in terms of developmental milestones related to specialization. Results indicated the proportion of athletes classified as specializers varied depending on the method used and athletes generally met specialization milestones after the age of 12. Study 2 examined the proportions of athletes who achieved ‘elite’, ‘pre-elite’, and ‘non-elite’ status in adulthood who were early specializers as determined by different methods. Results showed the method used changed the proportion of athletes classified as specializers at each level and there was no clear advantage or disadvantage to being a specializer. Combined, these studies provide intriguing data regarding the implications of different measures for assessing specialization in young athletes.

## 1. Introduction

Early specialization has been defined many ways, with little consistency between studies [1]. Collectively, however, these studies generally indicate early specialization involves dedicating large amounts of time and effort to one sport from a young age in pursuit of becoming an elite athlete, e.g., [2,3,4,5]. Precise determination of ‘young age’ and ‘early’ have yet to be established across the field [6]; however, the Developmental Model of Sport Participation (DMSP) suggests 12 years of age or earlier as a critical cut-off point [2].

The number of athletes following the path of early specialization has appeared to have seen an increase in recent years, arguably due to the professionalization of youth sport [7]. While there is much debate about the potential negative outcomes associated with *early* specialization, the need to eventually specialize is strongly supported [8].

Much of the theoretical rationale for early specialization is found in the deliberate practice framework [9]. Ericsson and colleagues used data from musicians as the foundation for the framework, noting that expert musicians spend more time in highly focused, effortful practice aimed at improving performance compared to their lesser skilled peers. The authors suggested engagement in this specific type of training (i.e., activities that are not inherently enjoyable, designed for the purpose of improving performance and not instantly gratifying, which they labeled ‘deliberate practice’) was the key mechanism explaining differences between those who achieve expertise and those who do not. Essentially, this framework is grounded in the notion that in order to become an expert, one must engage in a large quantity of deliberate practice; the greater the time spent in deliberate practice, the higher the attained level of performance. Importantly for our context, Ericsson and colleagues [9] suggested those who started deliberate practice at a later age (i.e., specialized later) were at a disadvantage compared to their peers who began earlier.

Despite the suggestions that early specialization in sport is increasing, recent studies have found later specialization (described as specialization after the age of 12) to be more common among elite athletes. The average age of specialization was 14 years of age in a study of elite hockey players, with only 12% of athletes specializing before age 12 [10]. Similarly, in a study of Olympic track and field athletes, 17 was the average age of specialization [11]. Despite these and other studies providing evidence against the necessity of specializing early, many parents and athletes still believe early specialization is the optimal way to become a top performing athlete. In a large study of 3090 athletes playing at high school, collegiate, and professional levels, 79.7%, 80.6%, and 61.7%, respectively, agreed that specializing in one sport helps an athlete play at a higher level [12]. Additionally, a study of over 900 youth athletes found 91% believed specialization in one sport increased their chances of getting better at their sport and 66% felt it would increase their chances of making a college team [13]. Unfortunately, early specialization is linked to an increased risk of injury and potential for burnout [4,14], which is why many sport organizations have advised against the practice (e.g., American Academy of Pediatrics) [7]. However, findings regarding early specialization should be interpreted with caution, as multiple authors, e.g., [12,15,16,17] have highlighted several methodological shortcomings.

For many years, sport researchers have warned about the lack of a clear and consistent definition of early specialization [12,15], while others have raised concerns around the validity of current methods used to classify athletes as ‘specializers’ [16,17]. A recent systematic review by Mosher et al. [1] found inconsistent definitions of early specialization across 48 empirical studies, with 18 different methods used to classify athletes as early specializers. ‘Single sport participation’ was the most common indicator of early specialization with ~73% of the 129 papers included in the review using this marker. Comparing single sport versus multi-sport athletes is one of the more common ways researchers have used to classify athletes as early specializers, e.g., [18]. Some have used a more comprehensive approach, collecting a complete history of an athlete’s sporting background, e.g., [19] while others have used a single yes/no item to determine specialization, e.g., [20]. A popular classification system in sports medicine is the “Sport Specialization Scale” developed by Jayanthi et al. [4], which classifies athletes on a spectrum from low to high specialization based on specialization as “year-round intensive training in a single sport at the exclusion of other sports” (p. 795). This scale, while used in youth populations (ages 7–18), does not include a measure of age and, therefore, does not distinguish between *early* specialization and specialization more generally. This variation in methods used to determine specialization can lead to inconsistent classification of who is a specializer. The scientific value of measuring specialization based on a dichotomized definition using arbitrary nominal variables needs to be evaluated and researchers are advocating for more adequate approaches (e.g., measuring a variety of continuous variables) [21]. However, as it is common practice to define and measure specialization in this way, there is value in examining the implications of different methods.

Importantly, the lack of a consensus definition can change the relationship and severity of outcomes associated with specialization. For example, in a study on the prevalence of specialization and injury history, Bell and colleagues [22] had high school students in the United States complete two different specialization classification tools, including a self-report as a ‘single sport’ or ‘multi-sport’ athlete, and the Sport Specialization Scale; both methods have been used in prior research to distinguish specializers from non-specializers. They found little agreement between the classification methods, with only 12% of students being classified as both single-sport and highly specialized and 26% being classified as multi-sport and low specialization. More troubling, the method used to classify athletes affected whether there was an association found between injury history and specialization. Athletes who self-classified as a single sport participant had no association to injury history, whereas those classified moderate or high specialization were more likely to report a history of injury [22].

The differing outcomes from different classification methods within the same study raise concerns about the reliability and validity of research examining specialization in general and early specialization in particular. While Bell and colleagues’ [22] study indicates specialization rates appear to be dependent on the classification method used, their study is only one among a rapidly growing research base focused on both early specialization and specialization more generally, with little consistency in the definition of the construct. Further, Bell and colleagues’ [22] study was cross-sectional in design, focused on athletes between the ages of 13 to 18. Given that much of the debate around specialization is concerned with *early* specialization and the dangers of specializing too soon, studies of this phenomenon in athletes before the age of 13 are needed. To this end, this investigation includes two studies, described below, examining the implications of varying classification methods for exploring both specialization and early specialization in sport.

## 2. Study 1

Given the recommendations from key organizations and athlete development models to avoid early specialization (e.g., American Academy of Pediatrics; Long Term Development Model) [7,23] coupled with issues related to approaches for classifying athletes as specializers, greater examination of early specialization measurement in youth (12 years of age and under) samples appears to be warranted. In this first study, we had two objectives. First, we examined how different approaches to defining and measuring specialization affected the classification of athletes throughout development. Based on prior work, our hypothesis was that the method used to determine specialization status would affect the proportion of the overall sample classified as ‘specializers’. Our second objective was to provide a ‘profile’ of the sample in terms of developmental milestones related to specialization. Few, if any, studies have provided individual milestones related to specialization, instead providing only the age at which specialization occurred. Examining the age that each component of specialization was met provides a more comprehensive picture of specialization patterns. Collectively, this study extends our understanding of the age at which youth meet different indicators of specialization and determines if *early* specialization is occurring in this sample.

### 2.1. Study 1 Materials and Methods

#### Participants

Participants included 362 athletes from one of the largest samples of athletes’ developmental histories [24,25]. In the original study, participants were recruited from all competition levels (e.g., local, regional, national, international) via advertisements on sport organization websites, social media and newsletters, or invitations from their coach. The sample was comprised of 203 females and 159 males with an age range of 14–42 (M = 20.8, *SD* = 4.7) from 10 different countries. The majority of participants were from Australia (*n* = 255) and Canada (*n* = 97). The athletes represented 36 different sports with the most popular being soccer (*n* = 77) and basketball (*n* = 46).

### 2.2. Measures

The indicators used in our studies were chosen mainly on their common use in the literature and not on their inherent value or evidence-based rationale. Data for this project came from a larger study of athlete development conducted in 2010–2011, where participants completed the Developmental History of Athletes Questionnaire (DHAQ) [26,27]. The DHAQ is a comprehensive instrument that gathers quantitative information on several different areas of an athlete’s history including their main sport practice history, other sport participation, family sporting history, and attainment of sporting milestones. The DHAQ has been previously validated [26] and used in samples with both able-bodied and para-athletes [24,28] While a number of measures included in the DHAQ were demonstrated to have questionable reliability or validity, the items analyzed for this study were deemed to have sufficient validity and reliability to provide valuable insights to the academic discussion of specialization in sport. A key strength of the DHAQ is that it allows collection and examination of key markers of specialization (outlined below) throughout athletes’ development.

#### 2.2.1. Single Sport Participation

As mentioned, single sport participation is the most commonly used indicator of specialization [1]. Subsequently, many studies have used this single qualifier as a method of classifying athletes as specializers [18,29]. While the merit of this method is debatable [22], due to its frequency of use, single sport participation was the first method used for comparison in this study. To operationalize this indicator, drawing from the DHAQ variables, we used the number of sports participated in at each age between 5 and 18 years.

#### 2.2.2. Year-Round Single Sport 

Another common indicator of specialization in previous work consists of year-round participation in one sport [1,20], and reflects a two-indicator method focusing on the ‘single sport participation’ captured in method one, described above, but within a fixed time frame (i.e., over the full year). In the current study, we consider these two indicators together as a method of classifying athletes as specializers. The questions/variables drawn from the DHAQ to operationalize this indicator were: (a) age of first participation in year-round training in main sport and (b) number of other sports participated in at each age (5–18).

#### 2.2.3. Sport Specialization Scale Items 

While the Sport Specialization Scale [4] was not explicitly used in the DHAQ, the three indicators that comprise the scale can be inferred from the data collected [26]. Specifically, the Sport Specialization Scale consists of three self-reported questions pertaining to whether an athlete is engaged in: (a) year-round training, (b) exclusion of other sports, and (c) participation in one main sport. The aligning questions/variables drawn from the DHAQ were: (a) age of first participation in year-round training in main sport, (b) age of deliberate exclusion of other sports, and (c) number of sports participated in at each age (5–18). On the Sport Specialization Scale, an athlete responds ‘yes’ or ‘no’ to each of the three questions; specialization is then scored as low, moderate, or high (i.e., ‘yes’ to one, two, or three indicators). While the three similar items of the SSS were used as indicators of early specialization, our analysis was somewhat different from how these data have been typically considered, as described below.

### 2.3. Coding

Information collected from the DHAQ was used to classify athletes as specializers at each age of development based on the three methods described above (i.e., single sport participation, year-round single sport, and the Sport Specialization Scale items). The proportion of athletes meeting the criteria of specialization based on each method at each age from 5 to 18 years was calculated. Unlike Jayanthi and colleagues’ [4] scale, which allowed for any one of the three indicators resulting in a score from one to three, this study used specific combinations of indicators (i.e., single sport alone; single sport and year-round training only; and single sport, year-round training, and exclusion of other sports). For a visual explanation of coding see Figure 1.

#### 2.3.1. Single Sport Participation 

The number of other sports in which an athlete participated was coded to indicate whether the athlete was a single sport participant and coded as 1 (i.e., response of zero additional sports) or participated in more than one sport (i.e., ≥1) coded as 2 for each age of development. In order to distinguish single-sport participation from data for years that participants had not yet started sport participation, athletes’ age of first participation in their main sport was established. If, for example, an athlete did not start participating in any sports until age seven, the zero response for other sport participation at age 5 and age 6 was coded as a 2 rather than a 1. Additionally, in order to distinguish single sport participation from those who could not fill out information because they were younger than the age of study (i.e., any age under 18), the age of the participant during the study was taken into consideration. This meant if an athlete was 15 years old at the time of the study, for the ages after 15 (16–18) a zero response for other sport participation was marked as ‘N/A’ rather than single sport participation. At each age of development, an athlete was deemed a specializer if they received a score of 1 on this measure and a non-specializer if they received a score of 2. 

#### 2.3.2. Year-Round Single Sport 

Year-round participation was coded similarly for each age of development. If the age of first participation in year-round training was equal to or less than the age being considered in the analysis (e.g., if the age of first participation in year-round training was 6 and the age being considered was 6 or higher), the athlete received a coding of 1. If the age of first participation in year-round training was greater than the age being considered for analysis (e.g., if age of first participation in year-round training was 7 and the age being considered in the analysis was 6), the athlete was coded as a 2. The same coding described above was applied to determine single sport participation. The two measures were then summed, and an athlete was deemed an early specializer if they received a total score of 2 (e.g., as it would indicate a coding of one for both criteria); they were classified a non-specializer if they received a score of 3 or 4, indicating they did not meet both criteria. 

#### 2.3.3. Sport Specialization Scale items 

The same coding of 1 or 2 was applied to age of deliberate exclusion of other sports at each age of development. Similar to year-round participation, if the age of deliberate exclusion of other sports was equal to or less than the age being considered (e.g., if the age being considered in the analysis was 8 and the age of deliberate exclusion of other sports was 8), the athlete received a coding of 1. If the age of deliberate exclusion of other sports was greater than the age being considered (e.g., if age being considered in the analysis was 8 and age of deliberate exclusion of other sports was 9) the athlete received a coding of 2. Year-round participation and single sport participation were coded as described above. After summing the three measures, an athlete was deemed a specializer if they received a total score of 3 (i.e., a coding of 1 for all three criteria).

### 2.4. Analyses

To address objective one, the proportions of athletes classified as specializers at each age were calculated using each of the methods described above. Objective two involved calculating the averages for the developmental milestones related to specialization. Averages were calculated for age of first participation in main sport, age of first participation in year-round training, and age of exclusion of other sports. 

## 3. Study 1 Results

Analysis revealed athletes’ average age of first participation in their main sport was 9.5 (*SD* = 5.1) years of age, the mean age of first participation in year-round training was 14.4 (*SD* = 3.9) years of age, and the mean age of exclusion of other sports was 15.1 (*SD* = 3.5) years of age. Only 16.3% of athletes began excluding participation in other sports by the age of 12, while 26.2% began year-round training by 12 years of age. 

As expected, the number of athletes classified as specializers varied depending on the method used. Across every age, single sport participation resulted in the highest percentage of specializers. There was a large difference (20% or greater) between the proportion of specializers based on single sport participation compared to year-round single sport or the Sport Specialization Scale items between the ages of 5 to 12 (early specialization). By age thirteen (specialization), the differences between groups ranged from 20% to 6% with the greatest degree of convergence occurring at 18 years of age. Between the ages of 5 and 12 (early), the proportion of athletes classified as early specializers remained low for the Sport Specialization Scale items and year-round single sport ranging from 0% to 12%, while the proportion of specializers based on single sport participation started at a higher percentage (19%) at 5 years of age and continued to increase (to 32%) up to 12 years of age. After age 13, there was an increase in the number of specializers based on year-round single sport and the Sport Specialization Scale items from 30% up to 46%, while the proportion of specializers for single sport participation fluctuated between 46% up to 54%. For a full break down of percentages by classification method, see Table 1.

## 4. Study 1 Discussion

Results from Study 1 highlight several implications for those studying early specialization. First, based on the average age of first participation in year-round training (~14 years), exclusion of other sports (~15 years) and the small percentage of athletes who met these milestones at 12 years of age or earlier, specialization (12 years of age or earlier) does not appear to be overly prevalent in this sample of athletes. This is in line with a study by Swindell et al. [30] that found approximately 17% of athletes had specialized at 12 years of age or earlier. While this is a meaningful minority and each athlete’s experience and safety is important, these results suggest either the number of children meeting criteria for early specialization is not as large as the rhetoric would suggest, e.g., [7,31,32] or those who specialized early are no longer in the system at the age of this sample of athletes.

Second, our hypothesis suggesting differences in proportions of specializers based on method used was generally well supported, given large discrepancies found; the more indicators used, the lower the proportion of athletes classified as specializers.

At every age, using single sport participation as the sole indicator resulted in the highest number of athletes classified as specializers. Such variation in how early specialization is determined at younger ages [1] raises concerns regarding the methods used to generate the evidence for such strong condemnation of the dangers of early specialization, e.g., [33]. The choice of method has clear implications for how study results are positioned in the discourse around early specialization. At six years of age, for example, less than one percent of athletes were participating in *year-round-training* at the exclusion of other sports, yet approximately 23% were participating in only one sport. Our understanding of the appropriate level of sport engagement for youth participating in sport at this age is very limited. Many children at six years of age may simply be beginning sport participation, and it seems reasonable to attempt one sport at a time, yet these athletes would be classified as specializers according to the method used in many studies. The results from this study indicate that a measure with more indicators may be more suitable to classify athletes—particularly younger child-athletes—as specializers.

That the proportion of athletes categorized as specializers changes so dramatically based on the method used raises questions about the conclusions drawn from the evidence base on early specialization, particularly given the range of definitions reported in this literature [1] For example, how does the method used affect conclusions about the value of early specialization for becoming an elite athlete? We explore this issue in study 2.

## 5. Study 2

As a commonly agreed upon principle of early specialization is that it is a pathway often followed in pursuit of becoming an elite athlete [2], it is important to examine the relationship between athletes’ sport development pathway (i.e., early specialization), and their attained skill level (i.e., elite). In this second study, we explored whether those who specialized in their youth became elite athletes in adulthood by examining the proportions of specializers (by each method used in Study 1) who achieved ‘elite’, ‘pre-elite’, and ‘non-elite’ status.

### 5.1. Study 2 Materials and Methods

#### 5.1.1. Participants 

A sub-sample of athletes from Study 1 was included in Study 2 (*n* = 237). Because this analysis focused on the highest level of skill attained, participants were limited to those *above* 18 years of age at the time of data collection. This age was chosen based on an assumption that generally, there is still room to improve and increase level of competition at age 16 or younger but by 18, if an athlete has not yet reached elite status, their chances of becoming elite decrease dramatically.

#### 5.1.2. Measures 

The same classification methods in Study 1 were used in Study 2 to determine specializers (i.e., single sport participation, year-round single sport, and the Sport Specialization Scale items). For single sport participation, athletes were either specializers meaning they met the single sport criterion (i.e., participated in one sport, sum of 1) or non-specializers (i.e., participated in more than one sport, sum of 2). For year-round single sport, athletes were considered specializers if they met both criteria (i.e., year-round and one sport, sum of 2) and non-specializers if they met one or none of the criteria (i.e., year-round or one sport or neither, sum of 3 or 4). Finally, in order to get a better understanding of the extremes of specialization compared to non-specialization, for the Sport Specialization Scale items, athletes had to meet all three criteria to be a specializer (i.e., year-round and exclusion of other sports and one main sport, sum of 3) but *none* of the criteria to be a non-specializer (as opposed to meeting some but not all criteria, sum of 6). This method provided a greater contrast than combining the remaining athletes (i.e., combining those who met one or two or none of the criteria) would have. Additionally, as there was overlap between all three methods (i.e., all use single sport, two use year-round), we felt much of the information would be provided by the other two methods and, therefore, the extreme comparisons for the Sport Specialization Scale items would be more valuable.

To determine highest level of competition and subsequently which skill group (elite, pre-elite, or non-elite) athletes belonged to, the milestones section of the DHAQ [24] was used. This section determines the age at which each athlete participated in different levels of competition (i.e., local, regional, national, international). Using the Athlete Development Triangle framework [24,34] athletes were categorized into three groups: (a) ‘*elite*’ athletes had competed at a senior international level, (b) ‘*pre-elite*’ athletes had competed at a junior international or senior national level, and (c) ‘*non-elite*’ athletes were all those who competed in remaining lower levels of competition. 

### 5.2. Analyses 

We determined the percentage of specializers across ages by each method described above who achieved elite, pre-elite, and non-elite status. The number of athletes (*n*) in each group varied across each age as the number of specializers varied, as described in Study 1. 

## 6. Study 2 Results 

Results indicated differences between the proportions of elite, pre-elite, and non-elite athletes classified as specializers across ages based on the method used. For a complete profile of percentages across the method used, see Table 2 and Figure 2. Across all skill levels, single sport participation resulted in the highest percentage of athletes classified as specializers for each age. The largest differences between percentages classified as specializers based on method used across skill level occurred in the earlier ages (i.e., 5–12), with the difference generally decreasing with age.

Using the criterion of single sport participation, at the ages of 5–7, there was a higher percentage of specializers who became elite compared to pre-elite and non-elite, whereas when using both year-round single sport and the Sport Specialization Scale items, between the ages of 6 and 13 there was a higher percentage of specializers who became non-elite and pre-elite compared to elite. By all three methods, there was a higher percentage of athletes who were specializers at 18 who became elite compared to non-elite. 

## 7. Study 2 Discussion

The assumptions underpinning the need to specialize are that it improves an athlete’s performance [13] and helps their chances of playing at a higher level [12]. The results of this study challenge aspects of these assumptions and highlight a range of implications. First, reinforcing the results from Study 1, there were large discrepancies between the proportion of athletes in each skill group based on the method used to classify athletes as specializers. As mentioned previously, single sport participation is not a nuanced measure of specialization status. However, even when using a more multi-dimensional measure (i.e., the Sport Specialization Scale items), the proportion of elite athletes versus pre-elite or non-elite was low until the age of fifteen. This suggests early specialization (i.e., specializing at 12 years of age or earlier) has limited benefit to performance and long-term elite attainment. This supports previous research by Wilhelm et al. [35] suggesting early sport specialization is *not a* requirement to compete at the most elite levels of sport. However, it is important to also recognize the large number of athletes who did go on to be elite, who had specialized before 18 (as measured by all three methods)—indicating that specialization in later adolescence is a common pathway to elite performance. More specifically, these results suggest specialization prior to 18 years of age may be required to become an elite athlete—but specialization does not need to or should not occur too early (i.e., not prior to 12–15 years). Further research is necessary to determine more precise optimal age(s) of specialization (and potential mitigating factors)—for athletes to reach top skill levels in adulthood.

## 8. General Discussion

Collectively, both studies demonstrate that single sport participation (i.e., the most commonly used indicator in prior research) resulted in the greatest proportion of the sample being classified as specializers. The large number of athletes classified as specializers based on the single sport participation method indicates there may be many athletes being classified as specializers who are not ‘true’ specializers (i.e., investing time and effort in one sport for the purpose of improving performance). There are many reasons a child may be participating in one sport, including parental time constraints, family financial constraints, and/or the child enjoying one sport over others. Simply asking the number of sports in which a child is participating fails to distinguish those who are deliberately choosing to play one sport to improve performance from those who participate in one sport for other reasons. The latter may be less of a concern as the child would likely be at a lower risk of injury associated with specializing for performance. Much of the increased injury risk associated with specialization is often attributed to the volume of training and overuse [36]. A child participating in one sport recreationally would likely not meet the volume of training that would warrant concern. In future work, if researchers choose to use single sport participation as a distinguishing factor it should be used in combination with other variables to ascertain why the athlete only participates in one sport in order to distinguish true specializers (i.e., the explicit decision to specialize to improve performance) from single sport athletes (e.g., those who may not be able to afford multiple sports or participate in low amounts in one sport at a recreational level). 

When other variables were considered in addition to single sport participation (e.g., year-round participation), there were few to no athletes at the younger ages (i.e., 5–12 years) meeting the criteria for specialization. Much of the caution around specialization concerns the notion that specializing too soon is dangerous to the physical health and well-being of the athlete [37]. Collectively, both studies indicate that specialization is not prevalent at younger ages, but rather, is more common beginning in the early adolescent years (i.e., 13–16). Further research should focus on specialization in early adolescence, given similar concerns may arise (e.g., related to physical maturation, psycho-social outcomes) despite specialization being deemed more acceptable at this stage of development. 

Our ultimate goal for this series of studies was to examine the varying approaches to measuring specialization in the same group of athletes. The inconsistency of classifications across different methods raises important issues about the validity and reliability of prior work on early specialization. This is particularly important given the value of research synthesis approaches (e.g., systematic reviews, meta-analyses) for generating patterns of evidence to reflect conclusions in a field. Many practitioners advise against specialization for youth athletes, but without a consistent classification method for specializers, the evidence behind these recommendations is unclear.

One important next step could be increasing the accuracy of the measure used to classify athletes by increasing the number of criteria an athlete has to meet. As we note above, there are potentially important elements of specialized engagement that are not captured in current approaches (e.g., reason for specialization). It should be noted, however, that increasing the number of criteria does not guarantee a more accurate measure. A recent study on the Sport Specialization Scale [16], for example, found that 30% of athletes were misclassified as moderately specialized, because they had only ever participated in one sport and, therefore, failed to meet the criteria of exclusion of other sports, when they should have been considered highly specialized. A strong rationale for item inclusion and consideration to the way new items are best measured are necessary when attempting to increase the accuracy of future measures.

A final consideration is the common dichotomizing of specialization. By dividing athletes into one or two groups, researchers may miss important information that could help elucidate the links between specialization and negative outcomes. For instance, some researchers have operationalized specialization as a simple yes or no question, such as “are you a specializer or not?” [20]. This method provides little information about the indicators chosen as key markers of specialization and, thus, makes understanding which aspect(s) of specialization is (are) most harmful or beneficial more difficult. The same issue applies when asking an athlete whether they are a single or multi-sport athlete. While this provides some information about the breadth of their participation, it does little to illuminate areas of potential concern. It would be more valuable to collect information on hours spent in each sport or months spent training to identify where overtraining may occur, rather than a dichotomized single versus multisport variable. While the Sport Specialization Scale [4] aimed to move away from this trend by creating degrees of specialization (low, medium, high), some have questioned the validity of the scale [16,17] as it does not account for volume of training and does not consider athletes who only ever played one sport. Overall, future research should move away from the simple single sport participation or dichotomous classification of early specialization and move towards a more comprehensive understanding of athletes’ full participation history. 

While both studies in this paper have several strengths, it is important to also discuss potential limitations. The data collected was retrospective not prospective in nature and, therefore, only captures the responses from ‘survivors’ in the sport system. This a common issue in athlete development research but it means it is possible those who dropped out from sport may have different participation trajectories that are not captured in this analysis. Moreover, the retrospective design also raises concerns regarding the accuracy of recall by participants, with participants attempting recall information from as far back as 30 years. Additionally, the specialization status of the sports was not considered; examining sports designed around early specialization (e.g., gymnastics, where peak performance can occur before 18 years for age) as well as later specializing sports (e.g., triathlon) is an important next step to gain more nuanced understanding of specialization. Moreover, our sample contained more elite athletes than pre-elite and non-elite athletes and, therefore, may not be generalizable to the average sporting population. Finally, development is best seen as a continuous process, which makes it hard to set ‘cutoffs’ for assessing development. We decided that having a cutoff would be more useful for analyses, and relevant for practitioners. Since the age of 18 is often used as an important marker of the transition from adolescent to adult, we felt this was an appropriate age to use as the cutoff in study two. 

Despite these limitations, our results suggest important implications of the use of different definitions/measures for specialization in research studies. We are not alone in raising these concerns. Recently, there has been some progress towards a consensus definition of specialization [38] although whether this definition will be widely used remains to be seen [21]. While a consistent definition of specialization would be a positive step forward, this will need to be followed by addressing the methodological concerns highlighted in this study. Until there is a clear and consistent definition and aligning method used to classify specialization, position statements outlining risks should be interpreted with caution. In particular, future research should better measure and distinguish ‘early’ specialization from ‘sport specialization’; this is essential when making specific and age-based recommendations. Despite the number of position statements and the passion of the rhetoric in this area, our understanding of the potential costs and benefits of early specialization is far from complete.

## Figures and Tables

**Figure 1 sports-10-00034-f001:**
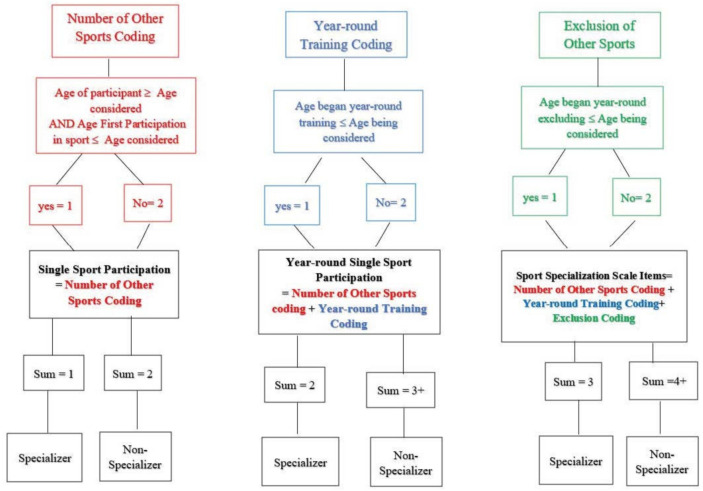
Breakdown of the coding for each variable and each method.

**Figure 2 sports-10-00034-f002:**
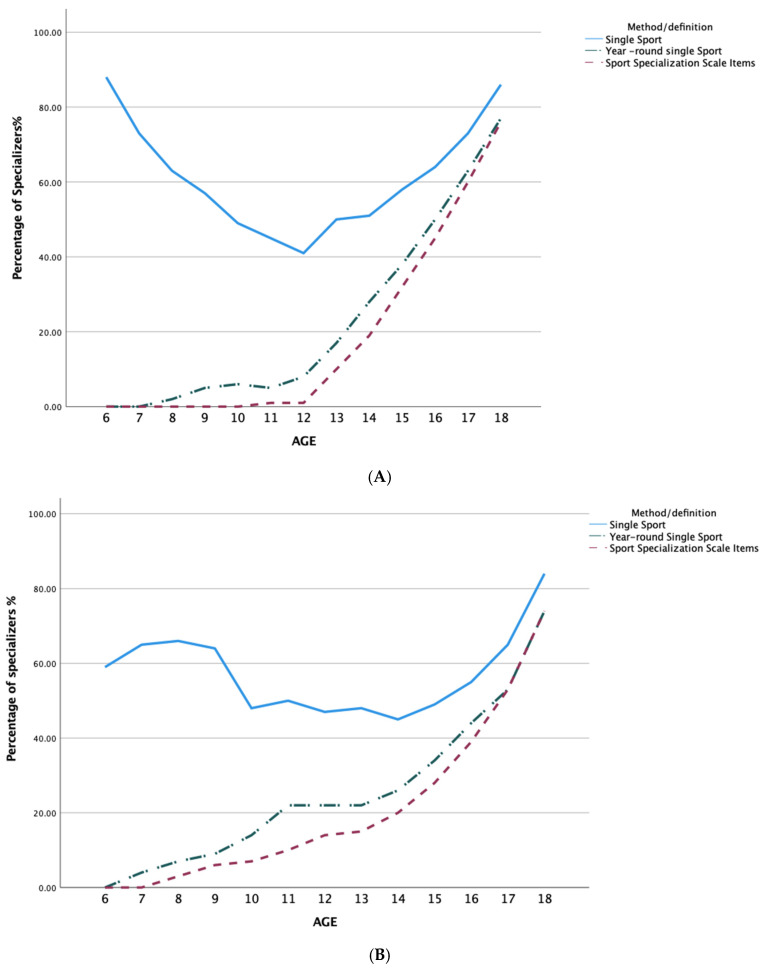
Percent of (**A**) Elite, (**B**) Pre-Elite, and (**C**) Non-Elite Defined as Specializers by Age and Method. Year-round single sport and the Sport Specialization Scale items have the same percentages and appear as one line.

**Table 1 sports-10-00034-t001:** Proportion of athletes classified as early specializers by method.

	Single Sport Participation	Year-Round Single Sport	Sport Specialization Scale Items
* Age *	% specializers (*n*)	% specializers (*n*)	% specializers (*n*)
Age 5	19.61 (71)	0.28 (1)	0.00 (0)
Age 6	23.48 (85)	0.83 (3)	0.55 (2)
Age 7	25.69 (93)	1.10 (4)	0.55 (2)
Age 8	29.01 (105)	2.21 (8)	0.83 (3)
Age 9	30.66 (111)	4.97 (18)	2.49 (9)
Age 10	30.66 (111)	7.73 (28)	3.59 (13)
Age 11	31.77 (115)	9.39 (34)	5.80 (21)
Age 12	32.87 (119)	12.43 (45)	7.73 (28)
Age 13	39.78 (144)	19.61 (71)	13.81 (50)
Age 14	45.86 (166)	30.11 (109)	25.14 (91)
Age 15	54.14 (196)	40.88 (148)	37.57 (136)
Age 16	**54.42 (197)**	45.58 (165)	42.54 (154)
Age 17	54.14 (196)	**46.13 (167)**	**45.30 (164)**
Age 18	50.83 (184)	44.48 (161)	44.20 (160)

Peak number of specializers indicated in bold.

**Table 2 sports-10-00034-t002:** Percentage of elite, pre-elite, and non-elite classified as specializers by method used.

	% Elite (*n*)	% Pre-Elite (*n*)	% Non-Elite (*n*)
* Age 5 *	*n* = 26	*n* = 13	*n* = 7
Single sport participation	88% (23)	77% (10)	71% (5)
Year-round single sport	n/a	n/a	n/a
SSS items	n/a	n/a	n/a
* Age 6 *	*n* = 33	*n* = 17	*n* = 10
Single sport participation	88% (29)	59% (10)	60% (6)
Year-round single sport	0% (0)	0% (0)	20% (2)
SSS	0% (0)	0% (0)	20% (2)
* Age 7 *	*n* = 41	*n* = 23	*n* = 11
Single sport participation	73% (30)	65% (15)	55% (6)
Year-round single sport	0% (0)	4% (1)	18% (2)
SSS items	0% (0)	0% (0)	18% (2)
* Age 8 *	*n* = 52	*n* = 29	*n* = 23
Single sport participation	63% (33)	66% (19)	54% (7)
Year-round single sport	2% (1)	7% (2)	15% (2)
SSS items	0% (0)	3% (1)	15% (2)
* Age 9 *	*n* = 61	*n* = 33	*n* = 15
Single sport participation	57% (35)	64% (21)	53% (8)
Year-round single sport	5% (3)	9% (3)	20% (3)
SSS items	0% (0)	6% (2)	20% (3)
* Age 10 *	*n* = 71	*n* = 42	*n* = 20
Single sport participation	49% (35)	48% (20)	50% (10)
Year-round one sport	6% (4)	14% (6)	15% (3)
SSS	0% (0)	7% (3)	15% (3)
* Age 11 *	*n* = 77	*n* = 46	*n* = 23
Single sport participation	45% (35)	50% (23)	52% (12)
Year-round single sport	5% (4)	22% (10)	13% (3)
SSS items	1% (1)	10% (5)	13% (3)
* Age 12 *	*n* = 85	*n* = 51	*n* = 27
Single sport participation	41% (35)	47% (24)	48% (13)
Year-round single sport	8% (7)	22% (11)	11% (3)
SSS items	1% (1)	14% (7)	11% (3)
* Age 13 *	*n* = 92	*n* = 58	*n* = 29
Single sport participation	50% (46)	48% (28)	41% (12)
Year-round single sport	17% (16)	22% (13)	10% (3)
SSS items	10% (9)	15% (9)	10% (3)
* Age 14 *	*n* = 102	*n* = 62	*n* = 33
Single sport participation	51% (53)	45% (28)	39% (13)
Year-round single sport	28% (29)	26% (16)	15% (5)
SSS items	19% (20)	20% (13)	15% (5)
* Age 15 *	*n* = 110	*n* = 65	*n* = 33
Single sport participation	58% (64)	49% (32)	36% (12)
Year-round single sport	38% (42)	34% (22)	18% (6)
SSS items	32% (35)	28% (18)	18% (6)
* Age 16 *	*n* = 114	*n* = 67	*n* = 33
Single sport participation	64% (73)	55% (37)	36% (12)
Year-round single sport	50% (58)	44% (30)	21% (7)
SSS items	45% (51)	39% (26)	21% (7)
* Age 17 *	*n* = 116	*n* = 68	*n* = 34
Single sport participation	73% (85)	65% (44)	47% (16)
Year-round single sport	63% (73)	53% (36)	26% (9)
SSS items	60% (70)	53% (36)	26% (9)
* Age 18 *	*n* = 119	*n* = 69	*n* = 34
Single sport participation	86% (102)	84% (58)	70% (24)
Year-round single sport	77% (92)	74% (51)	53% (18)
SSS items	76% (91)	74% (51)	53% (18)

## Data Availability

Data available upon request.

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
