# Peer review of "Proportions of Early Specializers Varies According to Methods and Skill Level"

_sports, 2022, doi:10.3390/sports10030034_

Round 1

Reviewer 1 Report

This is an important topic that the authors took under consideration in their study, so thank you for the opportunity of reviewing this paper. 

Below please find some comments and suggestions that could help to enhance and extend the scope of the article. 

Title and key words are ok in line with the contents of the article. 

Abstract is informative and concise.

Introduction is well-written but to broaden the context and strengthen the rationale for the study perhaps you could also look into some other issues like - the date of birth and sport achievements (for example "Physical and anthropological characteristics do not differ according to birth year quartile  in high-level junior Australian football players - Sports ) or team influence on the individual sporting success (for example "Multivariate analysis of the success factors in high-level male volleyball: a longitudinal study" Trends in Sport Sciences)  and perhaps also look up into gender issues - Do female athletes (or feminine sports ?)  have different trajectories and proportions of early specializers ? (Also look up "Contrasts in fitness, motor competence and physical activity among children involved in single or multiple sports" and "Fitness profiles of professional futsal players: identifying age-related differences from Biomedical Human Kinetics)

I think you could also bring up the problems of drop-out from sports due to too early specialization (look for example into: Sport Specialization, Part I: Does Early Sports Specialization Increase Negative Outcomes and Reduce the Opportunity for Success in Young Athletes? Sports Health. 2015) especially that you mention in the final paragraph of the Introduction that this kind of research is needed.

In the Methods section all is described thoroughly with details but I would expect some reliability and validity co-efficients cites where the research tools are described. This would enhance the reliability of the findings. I also had some concerns how were the participants selected - I know you have described the procedure but weren't there any missing links? missing individuals from niche sports for example?  

First study is nicely correlated with Study 2 and the transition is well-done and clear for the reader indicating why was the study undertaken, but I had problem with accepting the categorical age cut-off moment of 18 years 

(please look into Gulbin, Jason & Oldenziel, K.E. & Weissensteiner, Juanita & Gagné, Françoys. (2010). A look through the rear view mirror: Developmental experiences and insights of high performance athletes. Talent Development and Excellence. 2. 149-164 - maybe this will help with strengthening the rationale for your arbitrary decision).

Results are generally presented in clear manner, but I has a problem with fig.2 picture 3 - there are only two lines - where is the third line? 

The last part - General Discussion - I would suggest changing the subtitle - maybe into Summarizing the findings? or change the previous subtitles "Discussion" into something else as for now there is too much of these discussions. 

And concerning conclusions and your recommendations for the future actions in this area - I am not sure whether expecting one consensus definition isn't utopian thinking - some sports differ some much in terms of specialization (gymnastics vs rowing, weight lifting vs long distance running) that I am not sure any consensus can be worked out. Maybe phases of specialization should be more related to specific sport (or groups of sports? like team sports, water sports, coordination sports?)

Generally, this is a valuable input and with some enhancement should be proceeded further.

Discussion 

Reviewer 2 Report

Dear Authors,

I have attached a pdf file including my comments.

Best wishes,

Reviewer
